# Candidate Genes Encoding Dopamine Receptors as Predictors of the Risk of Antipsychotic-Induced Parkinsonism and Tardive Dyskinesia in Schizophrenic Patients

**DOI:** 10.3390/biomedicines9080879

**Published:** 2021-07-23

**Authors:** Elena E. Vaiman, Natalia A. Shnayder, Maxim A. Novitsky, Vera S. Dobrodeeva, Polina S. Goncharova, Elena N. Bochanova, Margarita R. Sapronova, Tatiana E. Popova, Alexey A. Tappakhov, Regina F. Nasyrova

**Affiliations:** 1Centre of Personalized Psychiatry and Neurology, V.M. Bekhterev National Medical Research Centre for Psychiatry and Neurology, 192019 Saint-Petersburg, Russia; naschnaider@yandex.ru (N.A.S.); scorpiona188@yandex.ru (M.A.N.); dobro.vera@gmail.com (V.S.D.); po.gon4arova@yandex.ru (P.S.G.); 2Centre for Collective Usage “Molecular and Cell Technologies”, V.F. Voino-Yasenetsky Krasnoyarsk State Medical University, 660022 Krasnoyarsk, Russia; 3Department of Pharmacology and Pharmaceutical Consulting, V.F. Voino-Yasenetsky Krasnoyarsk State Medical University, 660022 Krasnoyarsk, Russia; bochanova@list.ru; 4Department of Medical Genetics and Clinical Neurophysiology, Institute of Postgraduate Education, V.F. Voino-Yasenetsky Krasnoyarsk State Medical University, 660022 Krasnoyarsk, Russia; sapronova.mr@yandex.ru; 5Yakutsk Scientific Center for Complex Medical Problems, Department of Epidemiology of Chronic Non-Inflectional Diseases, 677018 Yakutsk, Russia; tata2504@yandex.ru; 6Department of Neurology and Psychiatry, Medical Institute, M.K. Ammosov North-Eastern Federal University, 677000 Yakutsk, Russia; dralex89@mail.ru

**Keywords:** extrapyramidal syndrome, antipsychotics, adverse drug reaction, antipsychotic-induced parkinsonism, antipsychotic-induced tardive dyskinesia, genetics, dopamine receptor, *DRD1*, *DRD2*, *DRD3*

## Abstract

(1) Introduction: Extrapyramidal disorders form the so-called extrapyramidal syndrome (EPS), which is characterized by the occurrence of motor disorders as a result of damage to the basal ganglia and the subcortical-thalamic connections. Often, this syndrome develops while taking medications, in particular antipsychotics (APs). (2) Purpose: To review studies of candidate genes encoding dopamine receptors as genetic predictors of development of AP-induced parkinsonism (AIP) and AP-induced tardive dyskinesia (AITD) in patients with schizophrenia. (3) Materials and Methods: A search was carried out for publications of PubMed, Web of Science, Springer, and e-Library databases by keywords and their combinations over the last 10 years. In addition, the review includes earlier publications of historical interest. Despite extensive searches of these commonly used databases and search terms, it cannot be ruled out that some publications were possibly missed. (4) Results: The review considers candidate genes encoding dopamine receptors involved in pharmacodynamics, including genes *DRD1*, *DRD2*, *DRD3*, and *DRD4*. We analyzed 18 genome-wide studies examining 37 genetic variations, including single nucleotide variants (SNVs)/polymorphisms of four candidate genes involved in the development of AIP and AITD in patients with schizophrenia. Among such a set of obtained results, only 14 positive associations were revealed: rs1799732 (141CIns/Del), rs1800497 (C/T), rs6275 (C/T), rs6275 (C/T) *DRD2*; rs167771 (G/A) *DRD3* with AIP and rs4532 (A/G) *DRD1*, rs6277 (C/T), rs6275 (C/T), rs1800497 (C/T), rs1079597 (A/G), rs1799732 (141CIns/Del), rs1045280 (C/G) *DRD2*, rs6280 (C/T), rs905568 (C/G) *DRD3* with AITD. However, at present, it should be recognized that there is no final or unique decision on the leading role of any particular SNVs/polymorphisms in the development of AIP and AITD. (5) Conclusion: Disclosure of genetic predictors of the development of AIP and AITD, as the most common neurological adverse drug reactions (ADRs) in the treatment of patients with psychiatric disorders, may provide a key to the development of a strategy for personalized prevention and treatment of the considered complication of AP therapy for schizophrenia in real clinical practice.

## 1. Introduction

Extrapyramidal disorders are so-called extrapyramidal syndromes (EPS), which are characterized by the occurrence of motor disorders as a result of damage to the basal ganglia and the subcortical-thalamic connections. Often, this syndrome develops while taking medications, in particular antipsychotics (APs). One such extrapyramidal disorder is drug-induced parkinsonism. This is an adverse reaction (ADR) from the extrapyramidal system that occurs while taking medications, most often APs in patients with schizophrenia, which belongs to the group of secondary parkinsonism with an average global prevalence of 36% [1,2,3,4,5]. Clinically, antipsychotic-induced parkinsonism (AIP) is characterized by the appearance of akinetic-rigid syndrome with the presence of a typical triad (akinesia, bradykinesia, tremor), which characterizes the presence of parkinsonism syndrome in the patient. Mainly, the difference between AIP and Parkinson’s disease (PD), the symptoms of which are similar, is a different etiological factor—the intake of drugs that affect the production of dopamine, in particular AP [1]. Another current SE from the extrapyramidal system is AP-induced tardive dyskinesia (AITD), which is characterized by involuntary non-rhythmic choreiform or athetoid movements that occur during AP administration or within 4 weeks after their cancellation and which persist for at least 4 weeks from the debut of the AITD. The prevalence of AITD varies from 0.57% to 50%, and the peak falls on the Russian Federation according to the literature review [6].

There are many theories of the mechanisms of development of AIP (Table 1) [7] and AITD (Table 2) [7,8].

The theory of dopamine receptor blockade is dominant in development of AIP and AITD. However, the role of genetic risk factors was known for 40 years. In addition to environmental risk factors, genetic factors can contribute to individual differences in susceptibility to the development of AIP and AITD in patients with schizophrenia [41,42,43,44]. In recent years, numerous associative genetic studies were carried out to search for candidate genes and single nucleotide variants (SNVs)/polymorphisms, which are predictors of the development of AIP and AITD.

The results of studies in this area are limited and contradictory, since some of them studied various forms of movement disorders caused by AP, while others considered AP-induced EPS as a single clinical manifestation of this ADR [45]. In spite of the fact that the correction of the ADR that developed during AP administration in patients with schizophrenia in the form of dose reduction or additional pharmacological treatment proceeds with a favorable outcome, AIP and AITD are the major reasons for poor adherence of patients with schizophrenia to regular and long-term APs administration, which increases the risk of disease recurrence, worsens its prognosis, and can worsen the quality of life of a patient with schizophrenia [46,47,48]. Identification and consideration in real clinical practice of genetic predictors of AIP [7] and AP-induced TD [7,49] can not only improve the current understanding of its pathophysiology these ADRSs and genetics predictors of mechanisms and metabolism of APs [7] (Figure 1) but also allow personalized prediction of the risk of AIP and AITD in schizophrenic patients at risk before starting APs therapy [50].

The aim of this study was to review the results of studies of candidate genes encoding dopamine receptors as genetic predictors of development of AIP and AITD in patients with schizophrenia.

## 2. Materials and Methods

A search was carried out for full-text publications in English and Russian databases (PubMed, Web of Science, Springer, e-Library) by keywords (extrapyramidal syndrome; antipsychotics; adverse drug reaction; antipsychotic-induced parkinsonism; antipsychotic-induced tardive dyskinesia; genetics; dopamine receptor; *DRD1*; *DRD2*; *DRD3*; *DRD4*) and their combinations over the last 10 years. In addition, the review included earlier publications of historical interest. Despite extensive searches of these commonly used databases and search terms, it cannot be ruled out that some publications were possibly missed. In total, we found 674 publications, of which 28 studies corresponded to the objectives of this review.

## 3. Results

We analyzed all works that met the objectives of this review, including genetic associative studies and genome wide associative studies (GWAS) that demonstrated both positive and negative results, which is important from scientific and practical points of view for planning large studies in the population of the Russian Federation characterized by ethnic and racial heterogeneity. The negative associative studies of SNVs/polymorphisms with AIP and AITD in the studied populations presented in Table 3 and Table 4, especially those of heterogeneous ethnicity and race, indicate low prospects for their further study and inclusion in the genetic screening panels for AIP and AITD in patients with schizophrenia. At the same time, associative studies with repeated positive results indicate the importance of their further study on the example of the Russian population for the subsequent creation of molecular genetic tools (methodology) for real clinical practice, including panels for deoxyribonucleic acid (DNA) profiling of schizophrenic patients receiving AP.

In this review, we made an attempt to generalize and systematize studies that studied candidate genes associated with the development of AIP and AITD in patients with schizophrenia (Table 3andTable 4).

### 3.1. Antipsychotic-Induced Parkinsonism

#### 3.1.1. *DRD2* Gene

The dopamine D2 receptor is a G-protein coupled receptor located on postsynaptic dopaminergic neurons that is centrally involved in the mesocorticolimbic pathways [63]. Additionally, D2 receptors are known targets of AP, which are used to treat schizophrenia [64]. Results of research by Al Hadithy A.F. et al. (2008) showed that the carriage of the 141C Del allele of the *DRD2* gene encoding D2 receptors is associated with a 9.5-fold greater risk of developing AIP in patients receiving AP in comparison with non-carriers (*p* = 0.005). Moreover, after gender stratification, the relationship between the carriage of the 141C Del allele and muscle rigidity remained statistically significant in men (*p* = 0.0039) but not in women in the African population [45].

According to Dolzan V. et al. (2007), the carriage of Ins-141CDel and Ser311Cys polymorphisms of the *DRD2* gene was not associated with the risk of developing AIP according to the Simpson-Angus Scale (SAS) in the Slovenian population of 151 patients [65].

Güzey C. et al. (2007) showed the frequency of carriage of the A allele of the *DRD2* gene (Taq1A) was statistically significantly higher compared to the control group (*p* = 0.04), which confirms that the carriage of the A allele is associated with the risk of developing AIP in patients with schizophrenia [52].

Bakker P.R. et al. (2012) demonstrated that the SNV rs6275 of the *DRD2* gene was associated with the risk of AP-induced resting tremor (*p* = 0.0140) in schizophrenic patients tested using the Abnormal Involuntary Movement Scale (AIMS) and the Unified Parkinson’s Disease Rating Scale (UPDRS) [66]. However, after the Simes correction for multiple trials, no statistically significant associations were found. According to the authors, this may be due to the small sample size (n = 209) [41]. Knol W. et al. (2013), who studied the role of the carriage of 141CIns/Del and C957T polymorphisms of the *DRD2* gene, did not find their significant association with the risk of AIP development while taking haloperidol in patients with schizophrenia [42].

A study by Gunes A. et al. (2007) found no statistically significant associations between the carriage of Taq1A1, 311Cys, and -141CDel polymorphisms of the *DRD2* gene with the risk of developing AIP in comparison with groups of schizophrenic patients with and without AIP on the perphenazine therapy. To assess the severity of EPS, the authors used the SAS and the Barnes Akathisia Scale (BARS) [51].

According to the research results of Bakker P.R. et al. (2012), who analyzed the prognostic role of rs1800497 (A2A1 (=C/T)), rs6277 (T/C), rs6275 (C/T), rs1801028 (Ser/Cys (=C/G)), rs1076560 (C/A), and rs1799732 (CDel) of the *DRD2* gene, no statistically significant associations were found with the risk of developing AIP [41]. Comparable results were obtained by Koning J.P. et al. (2011), who did not reveal statistically significant associations between rs1800497 (TaqI_A) (C/T), rs6277 (C957T) (T/C), rs1800498 (TaqI_D) (T/C), and rs1799732 (−141C) (C/Del) of the *DRD2* gene and the risk of developing AIP in patients with schizophrenia. The severity of AIP in this study was assessed using the UPDRS scale in patients on AP therapy for a duration of at least 1 month [43].

Greenbaum L. et al. (2015) studied 21 SNVs associated with AIP, however, only carriage of SNV rs1800497 of the *DRD2* gene was associated with the risk of developing AIP in a sample of 390 Italian patients. However, these data were not confirmed in a study in a Jewish population of 203 patients [67].

#### 3.1.2. *DRD3* Gene

The dopamine receptor D3 is both an autoreceptor and a postsynaptic receptor. It is localized in the neurons of the limbic system of the brain, the function of which is associated with cognitive, emotional, and endocrine functions. There is also an opinion that the D3 receptor appears to mediate some of the effects of APs and drugs used in the treatment of PD, which were previously thought to interact only with D2 receptors [53].

Güzey C. et al. (2007) found no association between the risk of developing AIP and the carriage of the Msc1 polymorphism of the *DRD3* gene encoding the dopamine D3 receptor [52]. Knol W. et al. (2013) investigated the role of the Ser9Gly polymorphism of the *DRD3* gene but did not find statistical associations with the risk of developing AIP when assessed by the SAS scale while taking haloperidol in patients with schizophrenia [42]. Gunes A. et al. (2007) also did not find statistically significant associations between the carriage of this polymorphism and the risk of developing AIP when assessed using the SAS and the BARS scales and when comparing groups of schizophrenic patients with and without AIP on the perphenazine therapy [51].

In the work of Gassó P. et al. (2011), sequencing of the *DRD3* gene coding region was carried out, the results of which revealed the location of five SNVs: the well-known nonsynonymous SNV rs6280 (Ser9Gly) in exon 2; the synonymous SNV in exon 2 (rs3732783); the SNV rs324026 in intron 1; and the SNVs rs2134655 and rs9828406 in intron 4. However, none of the detected SNV were associated with the risk of developing risperidone-induced AIP when assessed by the SAS scale in patients with mental disorders compared with the control group [53]. However, in the previous study of these authors (2009), an association was found between the SNV rs167771 (G/A) of the *DRD3* gene and the risk of developing risperidone-associated AIP (*p* = 0.00010) [68].

A study by Koning J.P. et al. (2011) found no statistically significant associations of the SNV rs6280 (Ser9Gly; T/C) of the *DRD3* gene with the risk of developing AIP when assessed by the UPDRS scale in schizophrenic patients treated with APs for at least 1 month [43].

### 3.2. Antipsychotic-Induced Tardive Dyskinesia

#### 3.2.1. *DRD1* Gene

The D1 dopamine receptor (*DRD1*) gene was one of the first studied pharmacodynamic genes in relation to the risk of developing AITD, because all APs affect the dopamine system. According to a study by Lanning R.K. et al. (2016), the carriage of the G allele (rs4532) (A/G) of the *DRD1* gene was possibly associated with the risk of AITD in patients with schizophrenia [69].

According to a study by Lai I.C. et al. (2010), carriage of the GG genotype SNV rs4532 of the *DRD1* gene was associated with the risk of developing AITD according to the AIMS scale in 382 patients (*p* = 0.033). Additionally, according to the results of this study, the carriage of the CGC haplotype of rs5326-rs4532-rs265975 of the *DRD1* gene was also associated with the risk of developing AITD (*p* = 0.027) [57].

In a study by Dolzan V. et al. (2007), carriage of SNV A-48G of the *DRD1* gene was not associated with the risk of developing AITD according to the AIMS scale in a sample of 151 patients [65].

#### 3.2.2. *DRD2* Gene

The D2 dopamine receptor is mainly expressed in the basal ganglia, an area of the brain that regulates motor function [51]. Dopamine hypersensitivity is the mechanism underlying the development of symptoms of AITD in patients with schizophrenia [29]. It should be noted that the density of dopamine receptors type D2 is also higher in schizophrenic patients with AP-induced TD than in patients without it [64]. The familial occurrence of TD supports a genetic theory in the development of AITD [70].

Gene *DRD2* is located on chromosome 11q23.2. It remains one of the most promising candidate genes for schizophrenia and AITD. Müller D.J. et al. (2012) reported that the carriage of the C allele (rs1800497, TaqIA) is associated with the development of AITD in Taiwanese patients, which was confirmed by meta-analyses [71].

In a study by Funahashi Y. et al. (2019), the carriage of rs1799732 (-141C Ins/Del) of the *DRD2* gene was associated with the risk of development and the severity of AITD, determined using the AIMS scale. However, these results were not further confirmed by meta-analysis [59].

The polymorphism Ser311Cys (rs1801028) was analyzed in relation to the development of AITD in patients with permanent residence in East Asia, but most studies did not confirm these results. In the European population, an association of the carriage of the C allele (rs6277, C957T) with the development of AITD was found. However, in the Taiwanese population, an association of the carriage of the T allele (rs6275, C939T) was found. At the same time, J.P. Koning et al. (2012) found these two SNVs of the *DRD2* gene to be associated with AITD in Europeans [43].

In a study by Zai C.C. et al. (2018), the carriage of the polymorphism TaqIB (rs1079597) was associated with the risk of developing AITD [64]. At the same time, the carriage of the polymorphisms Taq1A (rs1800497) and 141CIns/Del (rs1799732) and the TC haplotype (rs6277) was involved in altering the expression of the *DRD2* gene. Other SNVs considered in this study, including rs1076560, rs2283265, rs2242591, rs2242593, and rs12364283, need further study to determine their role in the development of AITD [64].

Zai C.C. et al. (2007) analyzed 21 SNVs in their study. As a result, only carriage of SNVs C939T, C957T, and rs2242592 of the *DRD2* gene was associated with the risk of developing AITD in schizophrenic patients in a sample of 202 European Caucasian and 30 African-American subjects (*p* = 0.022, 0.013, 0.070, respectively) [72].

According to Segman R.H. et al. (2003), the carriage of Taq-I A, -141C Ins/Del, and Ser311Cys polymorphisms of the *DRD2* gene was not associated with the development of AITD in schizophrenic patients in the Jewish Ashkenazi population according to the AIMS scale [73].

In a study by Dolzan V. et al. (2007), carriage of Ins-141CDel and Ser311Cys polymorphisms of the *DRD2* gene was not associated with the risk of developing AITD according to the AIMS scale in a sample of 151 patients [64].

#### 3.2.3. *DRD3* Gene

SNV rs6280 (Ser9Gly) of the *DRD3* gene encoding the D3 type dopamine receptor is localized on chromosome 3q13.31, and it is the most studied in AITD, because a relationship between the glycine variant and the risk of AITD was found. Substitution of the ninth amino acid serine for glycine leads to an increase in the affinity of the dopamine receptor D3 for dopamine in brain cells [74].

Zai C.C. et al. (2018) found that the carriage of the G allele (rs905568) in the 5′ region of the *DRD3* gene was associated with the risk of development and the severity of AP-induced TD [49,64,75,76].

In a study by Steen V.M. et al. (1997), carriage of the Ser9Gly polymorphism of the *DRD3* gene was associated with the risk of developing AITD in 100 patients with schizophrenia according to the AIMS scale [77]. Similar results were obtained by Lerer B. et al. (2002) in a sample of 780 patients [78]. According to Basile V.S. et al. (1999), the carriage of the MscI polymorphism of the *DRD3* gene was associated with the risk of developing AITD according to the AIMS scale in 112 patients with schizophrenia (*p* < 0.0005) [79].

In a study by Segman R. et al. (1999), carriage of DRD3Ser-Gly and DRD3Gly-Gly polymorphisms was statistically significantly associated with the risk of developing AITD in 180 patients with schizophrenic according to the AIMS scale (*p* = 0.02) [80,81].

## 4. Discussion

In this work, we analyzed 28 studies conducted from 1989 to 2021, studying 37 genetic variations, including SNVs/polymorphisms of four candidate dopamine receptor genes (*DRD1*, *DRD2*, *DRD3*, *DRD4*) involved in the development of AIP and AITD in patients with schizophrenia. Among such a variety of results obtained, only 14 positive associations were identified (Figure 2).

However, at present, it should be recognized that there is no final or unique decision on the leading role of any particular SNVs/polymorphisms in the development of AIP and AITD. The number of associative genetic studies of AIP and AITD is increasing in the last decade, which indicates both the relevance of the problem we are considering and the need to plan large-scale associative studies on the example of a Russian population heterogeneous in ethnicity and race due to the fact that the overwhelming majority of studies with positive associations were held abroad.

On the other hand, there is no doubt of the importance of translating the results of molecular genetic studies into real clinical practice for predicting the risk of developing AIP and AITD in patients with schizophrenia [80]. At the same time, it is required to take into account, first of all, associative genetic studies of AIP and AITD with repeated positive results for the development of valid genetic panels and decision-making models when conducting medical genetic and pharmacogenetic counseling of patients suffering from schizophrenia, both with those with existing AIP and AITD at the time of referral to a neurologist/psychiatrist and in patients potentially at risk of developing AIP (family history of Parkinson’s syndrome) and AITD.

Identification of genetic biomarkers that can help predict the manifestations of movement disorders caused by APs are of great importance. Understanding the genetic predictors of AIP and AITD and applying this knowledge can provide important information about the pathogenesis and the pathophysiology of the neurological SE under consideration, provide new goals for the development of drugs for effective correction of AIP and AITD, and help in the search for biomarkers from the perspective of personalized psychiatry and neurology. Such ambiguous results may be due to differences in the design of the studies we analyzed, variability in the samples of patients with schizophrenia, epigenetic interactions, pleiotropia, differences in methods for assessing the severity of AIP and AITD (different scales: AIMS, SAS, BARS, UPDRS), and therapeutic experience [81].

Thus, some of the SNVs/polymorphisms that were not associated with any previously studied phenotype cannot be completely excluded for further analysis, because they may still play a role in other ethnic samples of schizophrenic patients. Moreover, there is no clear understanding of what the molecular basis of AIP and AITD might be. This incomplete biological knowledge reduces the clinical (prognostic) value of associative genetic studies. Moreover, several candidate genes can have an additive effect. All of them should be included in the subsequent theoretical and statistical analysis to clarify the genetic predictors of AIP and AITD. One of the possible ways to overcome this problem is to study the pathways of molecular cascades (and their genes) as a whole rather than individual variations [72,73].

We found two studies of GWAS of AIP and six GWAS of AITD, one of which was performed in animal models. However, we did not find any positive associations between candidate genes for dopamine prescriptions and the risk of developing AIP and AITD in humans (Table 5 and Table 6); these were found in the animal model (mice) of AITD only (Table 6) [82,83].

## 5. Conclusions

Disclosure of genetic predictors of AIP and AITD as the most common neurological ADRs in the treatment of patients with schizophrenia and other psychiatric disorders may provide a key to developing a strategy for personalized prevention and therapy in real clinical practice. Taking into account the carriage of SNVs/polymorphisms of the studied candidate dopamine receptor genes associated with a high risk of developing AIP and AITD, therapeutic strategies can be individually changed in each specific clinical case. However, it should be recognized that the question of the genetics of AIP and AITD is far from being resolved.

## Figures and Tables

**Figure 1 biomedicines-09-00879-f001:**
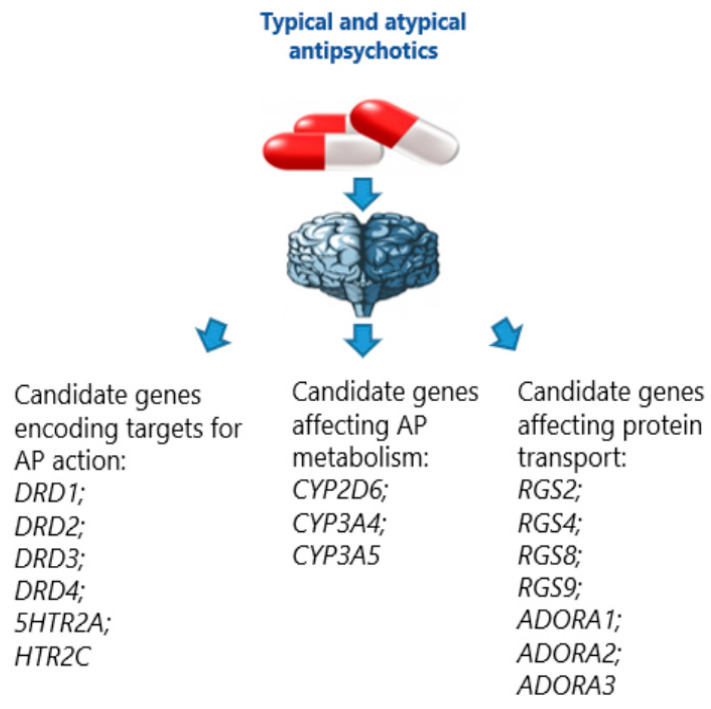
Candidate genes predisposed to development of antipsychotic-induced extrapyramidal syndrome.

**Figure 2 biomedicines-09-00879-f002:**
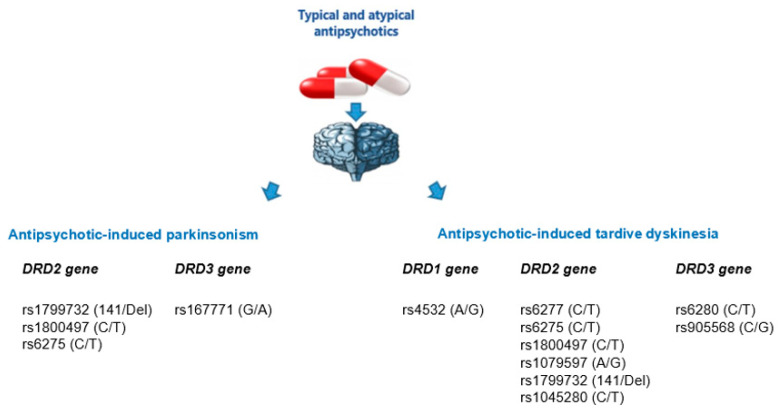
Positive associations of single nucleotide variants and polymorphisms of dopamine receptor genes with the risk of developing antipsychotic-induced parkinsonism and antipsychotic-induced tardive dyskinesia.

**Table 1 biomedicines-09-00879-t001:** Mechanisms of development of antipsychotic-induced parkinsonism [7].

Theory	Year	
Dopamine D2 receptor blockade	1983	[9]
Oversaturation (“clogging”) of type D2 striatal dopaminergic receptors	1988	[10]
Influence of the basal ganglia of the thalamocortical motor loop	2000	[11,12]
“Fast-off-D2” theory	2010	[13]
Role of adenosine receptors	2014	[14]
Blockade of the serotonergic system	2002	[15]
Cholinergic theory	2005	[16]
Melatonin theory	1983	[17]
Oxidative stress theory	1994	[18]
The role of vitamin D3	2009	[19]
Genetic theory	2011	[20]

**Table 2 biomedicines-09-00879-t002:** Mechanisms of development of antipsychotic-induced tardive dyskinesia [7].

Theory	Year	-
Effect on dopaminergic receptors	-	-
Changes in the activity of type D1 dopamine receptors	1983	[21]
Oversaturation (“clogging”) of type D2 striatal dophinergic receptors	1988	[22]
Increased affinity of dopamine D4 receptors	1993	[23]
Involvement of neural pathways	2002	[24]
Disruption of dopaminergic neurotransmission	2002	[25]
Dopamine hypersensitivity	2002	[26]
Effects on dopaminergic neurons	-	-
Excitotoxicity	1988	[27]
Toxic damage to dopaminergic neurons	1996	[28]
Effect on neurons of the basal ganglia	*-*	-
Effect on muscarinic, cholinergic receptors in the striatum	1974	[29]
Decreased activity of glutamic acid decarboxylase in substantia nigra	1988	[30]
Violation of the activity of the pathways of the basal gangliaChanges in the activity of striatal efferent pathways	2000	[31]
Increase/decrease of GABAergic neurons of the substantia nigra	2007	[32]
Other theories of antipsychotic-induced tardive dyskinesia	-	-
Activation of estrogen receptors	1981	[33]
Disruption of melatonin metabolism	1983	[17]
Disruption of the endogenous opioid system	1993	[34]
Oxidative stress	1994	[35]
Blockade of serotonergic 5-HT2 receptors	2002	[24]
Decreased pyridoxine levels	2005	[36]
Genetic predisposition	2011	[37]
Interaction of antipsychotics with a cerebral microelement–iron	2013	[38]
Carbonyl stress and immune inflammation	2018	[39]
Role of neurotrophic factor	2005	[40]

**Table 3 biomedicines-09-00879-t003:** Candidate genes encoding dopamine receptors and predisposed to development of antipsychotic-induced parkinsonism.

Gene	Protein	Locus	SNV	Effect of SNV	*p*-Criterion	Sample of Patients (*n*)	Ethnicity	Authors
*DRD2*	Dopamine receptor type 2	11q23.2	rs1799732 (141CIns/Del)	Associates with risk of muscle rigidity of men	*p* = 0.0039	126	Africans	[45]
				Not associated with the risk of developing AIP	*p* > 0.05	209	-	[41]
						150	-	[42]
						402	-	[43]
						47	-	[51]
			rs1800497 (C/T)	Polymorphism TaqIA carriage is associated with the risk of developing AIP	*p* = 0.04	119	Italians	[52]
				Not associated with the risk of developing AIP	*p* > 0.05	402	-	[43]
						47	-	[51]
						209	-	[41]
			rs6275 (C/T)	Associated with AP-induced resting tremor	*p* = 0.0140	209	Dutch	[41]
			rs1800498 (T/C)	Not associated with the risk of developing AIP	*p* > 0.05	402	-	[43]
			rs1076560 (C/A)			209	-	[41]
			rs6277 (T/C)					
						402	-	[43]
			rs6275 (C/T)			209	-	[41]
			rs1801028 (C/G)					
*DRD3*	Dopamine receptor type 3	3q13.31	rs167771 (G/A)	Associated with the risk of developing AIP	*p* = 0.00010	126	Italians	[53]
			rs6280 (T/C)	Not associated with the risk of developing AIP	*p* > 0.05	150	-	[42]
						47	-	[51]
						321	-	[53]
						402	-	[43]
			rs3732783 (T/C)			321		[53]
			rs324026 (C/T)					
			rs2134655 (A/G)					
			rs9828406 (A/G)					

**Table 4 biomedicines-09-00879-t004:** Candidate genes encoding dopamine receptors and predisposed to development of antipsychotic-induced tardive dyskinesia.

Gene	Protein	Locus	SNV	Effect of SNV	*p*-Criterion	Sample of Patients (*n*)	Ethnicity	Authors
*DRD1*	Dopamine receptor type 1	5q35.2	rs4532 (A/G)	Associated with the risk of developing AITD	*p* = 0.02	836	Asians	[54,55]
*DRD2*	Dopamine receptor type 2	11q23.2	rs6277 (C/T)	Associated with the risk of developing AITD (allele C)	*p* < 0.05	402	Dutch and Belgians	[43]
			rs6275 (C/T)	Associated with the risk of developing AITD (allele T)				
				Not associated with the risk of developing AITD	*p* > 0.05	263	-	[56]
			rs1800497 (C/T)	Polymorphism TaqIA carriage is associated with the risk of developing AITD	*p* < 0.05	206	Americans	[57]
				Not associated with the risk of developing AITD	*p* > 0.05	402	-	[43]
						263	-	[56]
			rs1079597 (A/G)	Polymorphism TaqIB carriage is associated with the risk of developing AITD	*p* < 0.05	369	Americans	[58]
			rs1799732 (141CIns/Del)	Associated with the risk of developing AITD	*p* < 0.001	100	Japaneese	[59]
					*p* = 0.001	402	Dutch and Belgians	[43]
			rs1800498 (T/C)	Not associated with the risk of developing AITD	*p* > 0.05	263	-	[56]
						402	-	[43]
			rs1801028 (C/G)			263	-	[56]
			rs1045280 (C/T)	Associated with the risk of developing AITD	*p * = 0.025	381	Chineese	[60]
*DRD3*	Dopamine receptor type 3	3q13.31	rs6280 (C/T)	Not associated with the risk of developing AITD	*p * = 0.021	216	Chinese	[61]
					*p* > 0.05	836	-	[55,58]
					*p* > 0.05	402	-	[43]
			rs905568 (C/G)	Associated with the risk of developing AITD	*p* < 0.05	171	Americans	[55,58]
			rs9817063 (T/C)	Not associated with the risk of developing AITD	*p* > 0.05	168	-	[56]
			rs2134655 (G/A)					
			rs963468 (G/A)					
			rs324035 (C/A)					
			rs3773678 (C/T)					
			rs167771 (A/G)					
			rs11721264 (G/A)					
			rs167770 (A/G)					
			rs9633291 (T/G)					
			rs1800828 (G/C)					
*DRD4*	Dopamine receptor type 4	11p15. 5	rs3758653 (T/C)					
			rs1800955 (T/C)			56	-	[62]

**Table 5 biomedicines-09-00879-t005:** GWAS of antipsychotic-induced parkinsonism.

GWAS	Genes in the Top of GWAS	Sample	Year	References
Human studies				
-	* ZFPM2 *	178	2011	[84]
CATIE	* EPF1 *, *NOVA1*, *FIGN*	3097	2009	[85]

**Table 6 biomedicines-09-00879-t006:** GWAS of tardive dyskinesia.

GWAS	Genes in the Top	Sample	Year	References
Human studies				
-	*FOXP1*	1103	2021	[86]
CATIE	* GSE1 *, *TNFRSF1B*, *CALCOCO1*, *EPB41L2*	1406	2021	[87]
-	* HSPG2 *	100	2011	[88]
-	* ZNF202 *	738	2010	[89]
-	* SLC6A11 *, *GABRB2*, *GABRG3*	100	2008	[90]
Animal models (mice)				
EMMA	*ZIC4*, *NKX6-1*, *GRIN1*, *GRIN2A*, *DRD1A*, *DRD2*	27	2011	[82]

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
