# Peer review of "Candidate Genes Encoding Dopamine Receptors as Predictors of the Risk of Antipsychotic-Induced Parkinsonism and Tardive Dyskinesia in Schizophrenic Patients"

_biomedicines, 2021, doi:10.3390/biomedicines9080879_

Round 1
Reviewer 1 Report
The review summarizes genetic studies on the association of dopamine receptor (DR) polymorphisms with EPS. The topic is of interest but a number of issues should be addressed:
Major concerns:
1) Clarity of the text would benefit from extensive editing for English language
2) The authors posit that genetic polymorphisms might be predictive and allow risk stratification for development of EPS prior to administration, guiding therapeutic choices. This is main rationale for writing the current review, yet several elements are absent that would enrich this discussion. For example, the authors do not discuss: 1) the relative strength of associations (RR, OR) for any given gene or SNP; 2) whether any of these loci are predictive on an individual level for AIP/AITD; 3) the roles of non-DR loci and how they might interact with DRDs to influence risk (separately, can the authors discuss why they limited their search to DRs only??); 4) how genetic risks might interact with other disease and demographic factors to influence individual risk
Minor concerns
1)Did the authors examine the reference lists of publications from their search for other relevant articles? This is a common approach when performing a literature search of this type.
2) In tables 1/2, could the authors provide citations for the publications supporting the hypotheses listed rather than just the year?
3) Line 80 “major” might be better than main. There are many factors that contribute to poor adherence
4) line 107 genome wide not genome whole for GWAS
5) typical and atypical are misspelled in figure 2
Author Response
Dear Reviewer,
We thank you very much for your assessment of our manuscript and recommendations for its revision.
We have tried to take into account all your comments and recommendations whenever possible.
Sincerely,
Authors

Reviewer 2 Report
This article summarizes recent researches on candidate genes encoding DA receptors as predictors of development of AIP and AITD in Schizophrenic patients. They analyzed 18 genome-wide studies and examined 37 genetic variations. Among such a set of obtained results, only 14 positive associations were revealed. They found that 14 gene variations are positively associated with AIP and AITD. Overall, it is a well-written article. Both positive and negative findings are included and balanced. Both the tables and figures are very helpful for readers to read and understand this article. I have only one comment.
There are two methods to identify the genetic variants underlying the inherited risk factors - candidate gene studies and GWAS. While GWAS have the power to identify novel risk genes for AIP and AITD, GWAS are still in the early stages and few genes identified by GWAS have been verified by clinical or preclinical studies. Here, I strongly encourage the authors to add one section to review recent progress in research on candidate genes in experimental animal models (if any). This is critically important in understanding the findings by GWAS.
Author Response
Dear Reviewer,
We thank you very much for your assessment of our manuscript and recommendations for its revision.
We have tried to take into account all your comments and recommendations whenever possible.
We found two studies of GWAS of AIP and six GWAS of AITD, one of which was performed in animal models. However, we did not find any positive associations between candidate genes for dopamine prescriptions and the risk of developing AIP and AITD in humans, but in animal model (mice) of AITD only. We have added two tables (Table 6 and Table 7) to the Discussion section.
Sincerely,
Authors

Round 2
Reviewer 1 Report
authors have adequately responded